# Autonomous Driving Decision Control Based on Improved Proximal Policy Optimization Algorithm

Qingpeng Song [1,2] , Yuansheng Liu [2,3,*] , Ming Lu [4] , Jun Zhang [3] , Han Qi [1,2] , Ziyu Wang [5] and Zijian Liu [2,3]

1 College of Smart City, Beijing Union University, Beijing 100101, China; 20201081210204@buu.edu.cn (Q.S.); 20211081210207@buu.edu.cn (H.Q.)
2 Beijing Key Laboratory of Information Service Engineering, Beijing Union University, Beijing 100101, China; 20201083510913@buu.edu.cn
3 College of Robotics, Beijing Union University, Beijing 100101, China; xxtzhangjun@buu.edu.cn
4 College of Applied Science and Technology, Beijing Union University, Beijing 100101, China; yykjtluming@buu.edu.cn
5 College of Urban Rail Transit and Logistics, Beijing Union University, Beijing 100101, China; wangziyu_zoey@163.com
* Correspondence: yuansheng@buu.edu.cn

**Abstract:** The decision-making control of autonomous driving in complex urban road environments is a difficult problem in the research of autonomous driving. In order to solve the problem of high dimensional state space and sparse reward in autonomous driving decision control in this environment, this paper proposed a Coordinated Convolution Multi-Reward Proximal Policy Optimization (CCMR-PPO). This method reduces the dimension of the bird's-eye view data through the coordinated convolution network and then fuses the processed data with the vehicle state data as the input of the algorithm to optimize the state space. The control commands acc (acc represents throttle and brake) and steer of the vehicle are used as the output of the algorithm.. Comprehensively considering the lateral error, safety distance, speed, and other factors of the vehicle, a multi-objective reward mechanism was designed to alleviate the sparse reward. Experiments on the CARLA simulation platform show that the proposed method can effectively increase the performance: compared with the PPO algorithm, the line crossed times are reduced by 24 %, and the number of tasks completed is increased by 54 %.

**Keywords:** deep learning; reinforcement learning; sparse reward environments; autonomous driving; decision and control

## 1. Introduction

Decision control of autonomous driving in urban environments is a challenge in the field of autonomous driving, due to the complex road geometry and the interaction of multiple agents. The realization of an intelligent decision control system that can handle complex road geometries and multi-agent interaction is essential for autonomous driving in urban environments. At present, the mainstream autonomous driving method divides it into three modules: perception, decision-making, and control, and the traditional perception module provides the vehicle with surrounding environment information through the vehicle's sensors, such as the vehicle's location information [1] and the vehicle's attitude information [2,3]. The perception system is an important part of autonomous driving, and a good perception system can provide reliable vehicle status data for decision-making and control, and then help the vehicle make the correct decision-making control. Traditional decision methods output decision results by dividing states and rules, such as finite state machines [4], decision trees [5], and so on. Traditional control methods can be divided into model-free control and model-based control, and the mainstream control methods include PID control, sliding mode control (SMC), and model predictive control (MPC) [6]. Vehicle

control is realized by tracking the path planned by the decision module [7]. The traditional decision control method has achieved good results to a certain extent, but it also has certain limitations—they need to artificially specify the driving scenario, so they only apply to known invariant scenarios; the artificial design rules do not fully satisfy the real needs of the scenarios, leading to a significant decrease in the applicability of the algorithms. With the application of machine learning methods in the field of urban intelligent transportation [8,9], the use of neural networks to process environmental information [10,11] and output vehicle control signals based on end-to-end imitation learning and reinforcement learning decision models has become a much-researched topic in decision control research [12]. The core of imitation learning is the construction of datasets of the driving behaviors of expert drivers for supervised imitation training [13]. Toromanoff [14] successfully achieved end-to-end lateral control of autonomous driving by labelling and training fisheye camera data. Geng's [15] proposed approach can plan out an optimized lane change path according to the vehicle condition by learning the excellent drivers' driving routes. However, imitation learning suffers from several weaknesses: it requires a large amount of data labelling; moreover, expert data cannot provide hazardous driving operations and therefore the trained system cannot make decisions about hazardous driving situations. Different from imitation learning, reinforcement learning does not require complex data annotation and only requires multiple trials and errors in the environment to obtain the optimal model. It combines the advantages of deep learning and has been widely used as deep reinforcement learning [16] to solve the decision and control problems of autonomous driving [17]. Park [18] proposes a path planning method for mobile robots based on deep deterministic policy gradient (DDPG) to overcome the sparse reward problem in autonomous driving mobile robots through hindsight experience replay technology. Kendall [19] used a monocular image as input and was able to learn lane-tracking strategies using only a small number of training scenes. Qiao [20] used an autonomous course-based deep reinforcement learning approach to solve simple urban intersection decision-making problems. Isele [21] implemented a Deep Q-network (DQN) [22] algorithm for driving decisions at intersections using the CARLA [23] simulation environment. Current research on deep reinforcement learning-based decision control for autonomous driving has made some achievements but the research work still exhibits shortcomings. Firstly, the algorithm needs to explore a lot during the training process, due to the high dimension of its input state data and the random policy sampled by the agent during the initial training, which makes it difficult to quickly obtain effective rewards [24]. Secondly, the design of the reward items is relatively single, so that the agent can only obtain the reward after interacting with the environment many times, and the reward in the intermediate process is difficult to evaluate. Finally, the methods DDPG [25], PPO [26], and A3C [27], which can output continuous actions, have stronger exploration capability and faster convergence speed compared to the DQN method with discrete space actions. The algorithm training process has a high cost of interaction with the environment. If it can solve the sparse reward problem to some extent and reduce the dimension of state space, the convergence of the model can be accelerated and the number of interactions with the environment can be reduced. The problems of high state space dimension include sparse reward and excessive number of interactions with the environment in deep reinforcement learning in autonomous driving decision control. Firstly, this paper used a bird's-eye view [28] as input, which provides abundant environmental information and has a low data dimension compared to a forward-looking camera or lidar data. To further extract the feature data to preserve its spatial information, this paper used the coordinate convolutional network [29] (Coordinate Convolution, CoordConv) to extract the feature data of the bird's-eye view. To provide spatial information for the algorithm, the feature data extracted above and the lateral distance between the vehicle and the lane line, the angle between the vehicle and the lane line, the current speed of the vehicle, and the distance to the vehicle ahead (collectively referred to as the vehicle state) were input to the algorithm for decision-making. Secondly, the proposed method designs a multi-objective reward mechanism for complex urban traffic scenarios according to the requirements of

vehicle safety, comfort, etc., effectively alleviating the problem of sparse reward. Finally, the CCMR-PPO algorithm controls the vehicle's acc and steer commands by making decision outputs from the reward values. Experiments show that the designed decision control model presented in this paper achieves a good performance in traffic scenarios with the participation of other vehicles, such as roundabouts and intersections.

## 2. Reinforcement Learning

The nature of reinforcement learning is interactive learning, i.e., it allows an agent to continuously interact with the environment to obtain the maximum cumulative reward value and thus learn the optimal strategy. The description of reinforcement learning is usually based on the well-known Markov decision process, as shown in Equation (1).

$$E = <S, A, P_{sa}^{s'}, P_0, R, \gamma>. \tag{1}$$

Six tuples are defined, where S and A are the set of states and the set of actions, $P_{sa}^{s'}$, denotes the probability of transferring from state $s \in S$ to state $s' \in S$ through a given action $a \in A$, $P_0$ denotes the initial state distribution, and $R$ is the reward function, which usually uses the cumulative discounted reward to define the state reward at moment $t$, $\gamma$ is the discount factor, as shown in Equation (2).

$$R = \sum_{i=t}^{T} \gamma^{(i-t)} r(s_i, a_i), \tag{2}$$

where the discount factor $\gamma \in [0, 1]$, whose smaller value indicates a greater focus on the current reward, and $r(s_i, a_i)$ indicates the value of the reward obtained by choosing action $a \in A$ in state $s \in S$. A policy $\pi$ is a mapping from states to action probability distributions: $\pi : S \rightarrow p(A = a|S)$. The goal of reinforcement learning is to learn an optimal policy $\pi^*$ that maximizes the expected cumulative reward for all states, i.e., the goal of reinforcement learning is as shown in Equation (3).

$$J = E_{s-p,a-\pi}[R_0]. \tag{3}$$

Under policy $\pi$, the value of state $s$ is denoted as $V^{\pi}(s)$, which represents the cumulative discounted reward brought by executing policy $\pi$ from state $s$. It is usually used as an actor in the actor-critic algorithm and is defined as shown in Equation (4).

$$V^{\pi}(s_t) = E_{s-p,a-\pi}[R_t|s_t]. \tag{4}$$

Similarly, under policy $\pi$, the value of action a taken for state $s$ is $Q^{\pi}(s, a)$, which represents the cumulative discounted reward brought by following policy $\pi$ after performing action a from state $s$. It is usually used as a critic in the actor-critic algorithm and is defined as shown in Equation (5).

$$Q^{\pi}(s_t, a_t) = V^{\pi}(s_t) E_{s-p,a-\pi}[R_t|s_t, a_t]. \tag{5}$$

In the policy gradient algorithm, a good policy is obtained by computing a policy gradient estimate and then using a stochastic gradient descent algorithm. The objective function of the network parameter $\theta$ update is as shown in Equation (6).

$$L(\theta) = E[\log \pi(a_t|s_t; \theta) A_t(s_t, a_t)]. \tag{6}$$

The update method for network parameters $\theta$ can be expressed as shown in Equation (7).

$$\theta_{t+1} = \theta_t + \alpha \Delta_\theta L(\theta_t), \tag{7}$$

where $A_t(s_t, a_t)$ is the estimate of the advantage function at moment t to measure the gap between the score obtained by the current policy interacting with the environment and the benchmark. The function is defined as shown in Equation (8).

$$A_t(s_t, a_t) = Q_t(s_t, a_t) - V(s_t). \tag{8}$$

In the PPO algorithm, a truncation method is usually used to limit the update of new policies with a loss function defined as shown in Equation (9).

$$L(\theta) = E[min(r_r(\theta)A_t, clip(r_r(\theta)), 1 - \epsilon, 1 + \epsilon)A_t], \tag{9}$$

where the ratio of the old to the new policy is denoted as shown in Equation (10).

$$r_t(\theta) = \frac{\pi_\theta(a_t|s_t)}{\pi_{\theta old}(a_t|s_t)}. \tag{10}$$

$\epsilon$ is the truncation constant, generally, with a value of 0.2; clip is the truncation function; and the value of $r_t(\theta)$ is limited to the range $1 - \epsilon$ and $1 + \epsilon$. In the process of parameter updating, the PPO algorithm uses truncation to limit the update of new policies, to avoid the problem of too-large differences in policies and improve the generalization of the algorithm.

## 3. Method

In this paper, the proposed CCMR-PPO method improved the implementation of the PPO. Its algorithm structure is shown in Figure 1, including three main modules: a CoorConv network, a multi-objective reward mechanism, and a PPO algorithm module. The CoorConv network extracts the effective information from the bird's-eye view and converts the $256 \times 256 \times 3$ bird's-eye view information into $256 \times 1$ low-dimensional data with key features. These feature data provide the CCMR-PPO with information about the location of other vehicles relative to the controlled vehicle. The multi-objective reward mechanism calculates the reward value according to the vehicle state data and passes the reward value into the experience pool of the PPO algorithm. The PPO algorithm inputs the fused state data into the policy network of the algorithm, generates the expectation and variance of the action through the current state, and then conducts random action sampling to generate commands for vehicle control. The evaluation network evaluates the performance of the policy network according to the data in the experience pool and then guides the policy network to update the parameters. When the new policy network is updated to a certain step, the old policy network is updated. Finally, the evaluation network and the policy network are updated periodically.

### 3.1. State Space

The state space contains the information required for autonomous driving decisions, including information about the vehicle's state and the road environment ahead. The positioning system is used to obtain the pose of the autonomous vehicle, a road route and map are obtained from a high-precision map, and the above information is transformed into a bird's-eye view. As shown in Figure 2a, the red rectangle is the controlled vehicle, always positioned in the lower middle region of the image, while the other vehicles are represented using green rectangles. Lanes are segmented using grey and white lines. The forward direction, relative position, speed, and lane information of these vehicles can be obtained from the figure. The blue area is the travel trajectory of the environmental plan. Figure 2b shows the front view image corresponding to the bird's-eye view. The controlled vehicle state space is defined as shown in Equation (11).

$$s = \{\tau, l, \phi, v, d\}, \tag{11}$$

where $\tau$ is the feature vector, obtained by inputting the bird's-eye view to the CoordConv network. $l$ is the lateral distance of the vehicle relative to the lane line, $\phi$ represents the angle between the vehicle and the lane line, $v$ is the current speed of the vehicle, and $d$ is the distance to the vehicle ahead. The above information is fused to obtain a new vector s as the input of the policy and evaluation network.

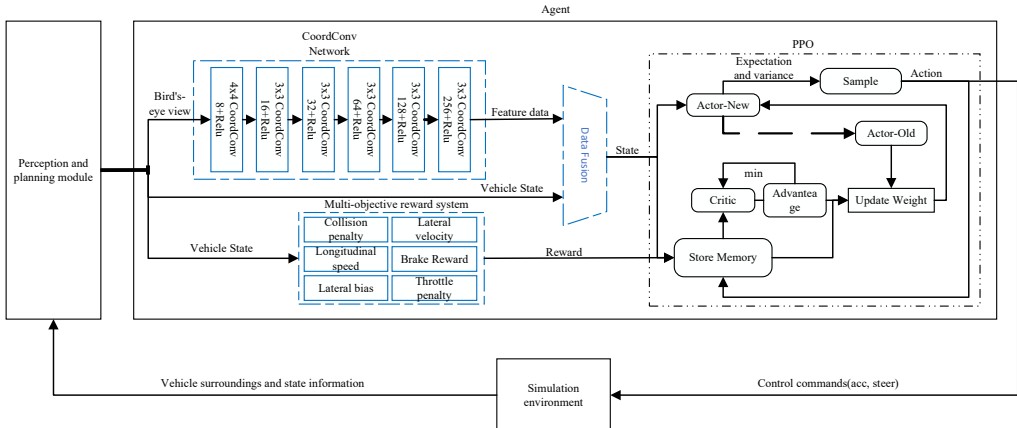

**Figure 1.** A framework for an autonomous driving agent. The agent takes information from the sensing module and generates a bird's-eye view, which is extracted by the CoordConv network into low-dimensional state data, which is then used by the PPO algorithm to learn policies to generate the correct control command.

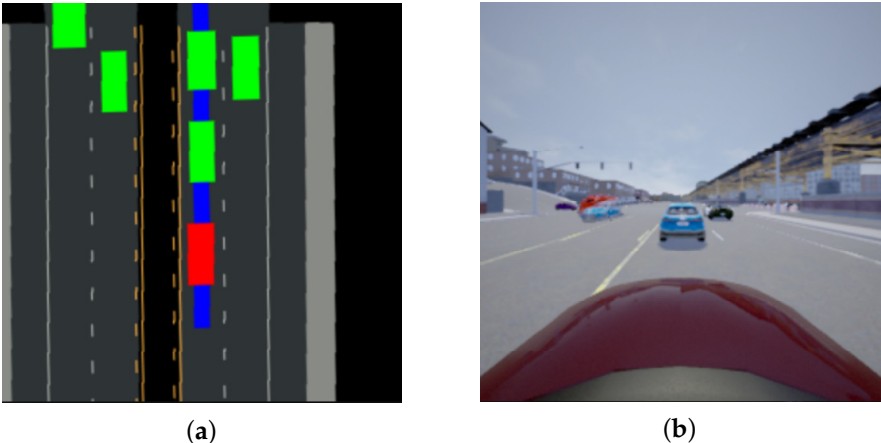

(**a**)             (**b**)

**Figure 2.** Simulation environment map: (**a**) Bird's-eye view, where the red box is agent and the green box is other vehicles. (**b**) Front view.

### 3.2. Action Space

The control of the vehicle is mainly divided into lateral control and longitudinal control. Longitudinal control is established by the longitudinal control parameter acc using the throttle and brake. The method normalizes *acc* to $[-1, 1]$, accelerates forward when *acc* is positive, and brakes when *acc* is negative. Lateral control is established by the steering wheel angle *steer* of the vehicle. The method normalizes *steer* to $[-1, 1]$, turns right when steer is positive, and turns left when steer is negative. So, the action space is defined as shown in Equation (12).

$$A = \{acc, steer\}. \tag{12}$$

### 3.3. Reward Function

In order to allow the agent to learn an excellent driving strategy, measure the quality of the actions performed by the agent, and solve the sparse reward problem of the algorithm, reference [30] considered a total of six environmental variables and state factors: collision

($r_{collision}$), longitudinal speed ($r_{speed\_lon}$,$r_{fast}$), lane departure($r_{out}$,$r_{line}$), lateral speed($r_{lat}$), brake($r_{brake}$), and throttle($r_{throttle}$). The last term $c$ is a small constant set to −0.1, which was used to penalize the ego vehicle for stopping still. The design of the reward function needs to encourage the vehicle to move forward along the lane while taking care that the change in action output is as smooth as possible. This paper split the targets according to their credit assignment, setting a scale of 50 parameter adjustment for the less frequent cases of the initial state such as the collision reward term $r_{collision}$, and a scale of 5 parameter adjustment for $r_{fast}$. For the initial frequently occurring cases such as rout, $r_{brake}$, and $r_{throttle}$, set a parameter adjustment with a scale of 1. For the other continuously changing parameters, $r_{speed\_lon}$, $r_{line}$ and $r_{lat}$, set a parameter adjustment with a scale of 0.1. Finally, this paper obtained the parameters in the paper and, after several tests, the reward function is as shown in Equation (13).

$$R = \begin{bmatrix} k_1 & k_2 & k_3 & k_4 & k_5 & k_6 & k_7 & k_8 \end{bmatrix} \begin{bmatrix} r_{collision} \\ r_{speed\_lon} \\ r_{fast} \\ r_{out} \\ r_{line} \\ r_{lat} \\ r_{brake} \\ r_{throttle} \end{bmatrix} + c \tag{13}$$

### 3.3.1. Collision Penalty

If the controlled vehicle collides with another vehicle or road boundary during network training, the current training episode ends and the next episode training begins. The collision penalty $r_{collision}$ is set to −1 if a collision occurs; otherwise, set to 0. With experimentation, it is verified that setting the weight of $k_1$ too small will result in frequent collisions, while if the weight of $k_1$ is set too large, negative reward values become common and the system cannot learn effective policy. Consequently, to ensure that the vehicle does not collide as much as possible, the factor $k_1$ of $r_{collision}$ is set to 200, as shown in Equation (14). This paper used 50 as a ruler to test the decision-making control effect of vehicles in dense and non-intensive traffic flow, and the experimental results show that the value of $k_1$ is set between 150 and 300, so the value of $k_1$ is set to 200.

$$r_{collision} = \begin{cases} -1, if\ collision \\ 0, otherwise. \end{cases} \tag{14}$$

### 3.3.2. Longitudinal Speed

$r_{speed\_lon}$ is the reward for the vehicle's longitudinal velocity $r_{speed\_lon}$. The greater the longitudinal speed, the greater the reward value obtained. The coefficient $k_2$ is set to 1. If the weight is set too large, the vehicle speed will increase rapidly and cause a collision. If it is set too small, the vehicle cannot move forward, as shown in Equation (15).

$$r_{speed\_lon} = v_{speed}. \tag{15}$$

In addition to a higher reward value for higher vehicle speeds, a certain penalty is set when the speed exceeds the threshold speed, where the penalty is denoted by $r_{fast}$. The threshold speed in this paper was set to $v_{thr}$ = 8 m/s, $r_{fast}$ was set to −1 when the vehicle speed was greater than 8 m/s, and 0 otherwise; its weight $k_3$ was set to 10, as shown in Equation (16).

$$r_{fast} = \begin{cases} -1, if\ v > v_{thr} \\ 0, otherwise. \end{cases} \tag{16}$$

### 3.3.3. Lane Departure

The lateral deviation of the vehicle is 0 when the vehicle is driving in the middle of its lane. When the vehicle is leftward, the value is negative, and the greater the deviation, the more negative the value. When the vehicle is rightward, the value is positive and, the greater the deviation, the larger the value. The lateral deviation penalty is set according to the properties of the function $e^{-|x|}$ as shown in Figure 3a. Equation (17) is used to constrain the vehicle to travel within the road, where $x$ is the lateral deviation of the vehicle from the lane line, and the reward value is maximum when the value of $x$ is zero and decreases exponentially with the increase of the deviation value. The coefficient $k_5$ is set to 1 and the characteristic curve of $r_l$ is shown in Figure 3b.

$$r_{line} = e^{-|x|^3+1}.$$ (17)

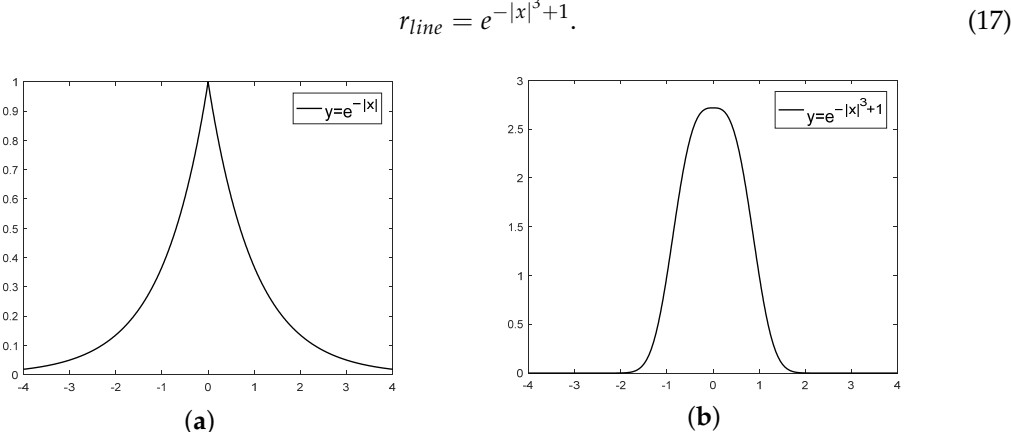

**Figure 3.** Negative exponential function characteristic graph. (**a**) Negative exponent. (**b**) Negative exponent of error.

This paper set the threshold value of the maximum lateral deviation of the vehicle as $l_{thr} > 2$ m; the training episode ends when the vehicle lateral deviation exceeds the threshold value, and the next episode is start. Experiments show that, if $l_{thr}$ is set too large, the vehicles are prone to collision and it is difficult to learn effective policy. if $l_{thr}$ is set too small, the vehicles have no room for trial and error, leading to early episode endings, and it is difficult to learn effective policy. The vehicle deviation penalty is defined as $r_{out}$, which is set to -1 when the lateral deviation of the vehicle exceeds $l_{thr}$ and to 0 otherwise. The coefficient $k_4$ is set to 5, as shown in Equation (18).

$$r_{out} = \begin{cases} -1, if\ l > l_{thr} \\ 0, otherwise. \end{cases}$$ (18)

### 3.3.4. Lateral Speed

To prevent the vehicle from driving unstably due to excessive lateral speed, the penalty $r_{lat}$ is set according to the steering wheel turning angle steer and speed $v_{speed}$. The coefficient $k_6$ is set to 0.2, as shown in Equation (19).

$$r_{lat} = -abs(steer) * v_{speed}^2.$$ (19)

### 3.3.5. Brake Reward

Braking should be applied when an obstacle is encountered in front or when the vehicle speed is greater than the speed threshold. The brake reward is $r_{brake}$. In this paper, the distance to detect the obstacle ahead was $d_{thr} = 15$ m. When there is an obstacle in front of the agent or the speed is greater than $v_{thr} = 8$ m/s, the braking value brake is used as the bonus value. The coefficient $k_7$ is set to 5, as shown in Equation (20).

$$r_{brake} = \begin{cases} brake, & if\ d < d_{thr}\ or\ v > v_{thr} \\ 0, & otherwise \end{cases}. \tag{20}$$

### 3.3.6. Throttle Penalty

Penalties are applied when the vehicle accelerates after the speed exceeds $v_{thr} = 8$ m/s. A throttle penalty is also applied to lateral deviations to prevent the vehicle from accelerating out of its lane in the event of a lane deviation. The throttle penalty is denoted as $r_{throttle}$. When the vehicle's speed is greater than $v_{thr} = 8$ m/s and the vehicle still accelerates or when the vehicle still accelerates after deviating from the lane, the throttle value throttle is used as the penalty value. The coefficient $k_8$ is set to 5, as shown in Equation (21).

$$r_{throttle} = \begin{cases} -throttle, & if\ v < v_{thr}\ or\ acceleration\ after\ crossin\ the\ line \\ 0, & otherwise. \end{cases} \tag{21}$$

### 3.4. Policy and Value Function as Neural Networks

This paper used the same actor-critic algorithm framework based on the design approach of the PPO algorithm, where the input state $S$ of the actor network gives the expectation $\mu$ and variance $\sigma$ of the next action. The expected value and variance of the actions are calculated based on the values of $\alpha$ and $\beta$ in the beta distribution and sampled to obtain the value of the action $a$. The critic network input state $S$ determines output state value function $V$ to evaluating the discounted reward of the policy. The structure of the two network models is the same, but the outputs are different. To keep the network updated synchronously and extract effective feature data, the weights of the previous CoordConv network are also updated at the same time. The specific structure of the CoordConv network is shown in Figure 1. There are six layers in the network, the size of the convolution kernel in the first layer is $4 \times 4$, and the size of the convolution kernel in the remaining layers is $3 \times 3$. The channels wide are 8, 16, 32, 64, 128, and 256, and the step size is 2. Relu is used as the activation function between the layers, and the data input is four frames of $256 \times 256$ bird's-eye view, and the output is $1 \times 256$ feature data. The actor network in this paper contained an input layer, two fully connected layers, and an output layer. The overall structure is shown in Table 1. The input layer contained 300 linear units, and the input data were the fused features extracted by CoordConv and the vehicle state. Two fully connected layers were designed with 100 linear units each, and the activation function of the first fully connected layer was tanh and the second was a soft plus. The output layer contained two linear units that output $\alpha$ and $\beta$. In the process of training the neural network, the Adam optimizer was used as the optimizer of the gradient loss function, and the learning rate took the value of 0.0005.

**Table 1.** Network structure of actor.

| Name of Network | Network Dimension | Activation Function |
|---|---|---|
| Input layer | 256 + 4 | / |
| Fully connected layer1 | 100 | tanh |
| Fully connected layer2 | 100 | soft plus |
| Output layer | 2 | / |

The critic network has the same general structure as the actor network, except that the activation function and the output layer have been changed. As shown in Table 2, the output layer of the critic network directly outputs the reward value with dimension 1. The input is the state information extracted from the experience pool. The experience pool of PPO stores the experience information of the previous h steps until the experience pool is filled, at which point it is sampled from the experience pool. The sampled state information is fed into the critic network to obtain the value function, and then the value function calculates the advantage function to update the actor network in reverse. In the

process of training the neural network, the same Adam optimizer is used as the optimizer of the gradient loss function, and the learning rate takes the value of 0.0005.

**Table 2.** Network structure of critic.

| Name of Network | Network Dimension | Activation Function |
| --- | --- | --- |
| Input layer | 256 + 4 | / |
| Fully connected layer1 | 100 | tanh |
| Fully connected layer2 | 100 | tanh |
| Output layer | 1 | / |

*3.5. Termination Conditions*

In the deep reinforcement learning process of interaction with the environment, there are fewer rewards obtained in the early stage, and the reward value cannot be increased, resulting in many negative memories stored in the experience pool, which affects the training speed, so the termination conditions need to be set; for the training reported here, the following termination conditions are set.

3.5.1. Specify the Number of Steps to Perform in the Episode

The maximum number of steps a vehicle can perform per episode in the environment is 1000. When its number of steps reaches 1000, the current episode is finished and the training for the next episode is started.

3.5.2. Out of Lane

The threshold of exceeding the distance from the lane is set to 2, which means that the vehicle exceeds the lateral range of 2 m, starting from the center of the lane. When the vehicle deviates from the threshold to the left or right, the current episode is ended and the training of the next episode begins.

3.5.3. Collision Occurs

When the vehicle collides with other vehicles or surrounding buildings, the current episode is ended and the training for the next episode is started.

**4. Results and Discussion**

*4.1. Experimental Environment*

The CPU model of the simulation hardware used in this study was Ryzen7 3700; the GPU model was RTX3060Ti; the memory stick had a capacity of 16G; and the operating system was Ubuntu 18.04. The simulation software environment was jointly built by CARLA and Python. The main parameters of CCMR-PPO were set as shown in Table 3.

**Table 3.** Main parameters of CCMR-PPO.

| Parameter | Value |
| --- | --- |
| Sampling time (s) | $10^{-3}$ |
| CoordConv network initial learning rate | $10^{-3}$ |
| Actor network initial learning rate | $5*10^{-3}$ |
| Critic network initial learning rate | $5*10^{-3}$ |
| Discount factor $\gamma$ | 0.99 |
| Clip parameter $\varepsilon$ | 0.2 |
| GAE parameter $\lambda$ | 0.95 |
| Batch size | 64 |
| Number of Episodes | 1600 |

All experiments in this paper were conducted in the CARLA simulator, and different driving scenarios were constructed by choosing the Town3 map in the simulator to train and

test the algorithm. The Town3 shown in Figure 4a contains a variety of urban road scenarios such as roundabouts, intersections, and curves. The map area size is 400 m × 400 m, where the total length of the road is 6 km, and the environment varies greatly from section to section. To improve the generalization ability of the model to allow vehicles to drive on different road sections in the environment and to ensure uniform distribution of samples, the locations at the beginning of each training were chosen randomly. At the same time, to increase the vehicle's ability to handle various traffic environment situations, other vehicles were added to the road traffic during each simulation, and the initial positions of these vehicles were set randomly. During the training process, the vehicle speed, steering, acceleration, coordinate attitude, and other data were obtained in real time through the program interface provided by CARLA, while an RGB camera and LIDAR sensors were added to the controlled vehicle to obtain environmental information. The RGB bird's-eye view of size 256 × 256 × 3 was obtained by transforming the above information and high-precision map projection.

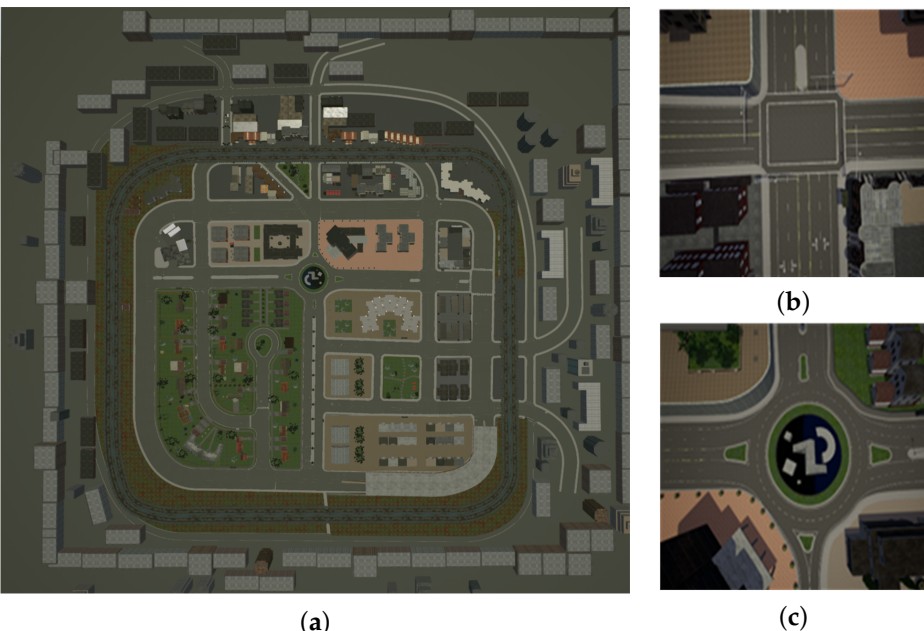

**Figure 4.** The simulation environment for algorithm training. (**a**) The map layout. (**b**) The crossroads. (**c**) The roundabout.

### 4.2. Comparative Analysis of Algorithms

To evaluate the performance of the algorithms in this paper, the original PPO, DDPG, and CCMR-DDPG, which incorporate CCMR ideas into the DDPG algorithm, were used as comparison algorithms. These algorithm models in this paper trained 1600 episodes, converged in about 300 episodes, and converged in 4.8 h. The state inputs of the original PPO and DDPG methods are bird's-eye views that are not processed by the CoordConv network, and the reward term was designed with reference to paper [30]. In the paper [30], four reward items $r_{speed\_lon}$, $r_{lat}$, $r_{collision}$, and $r_{out}$ were set, which represent speed, steering, collision, and exceeding lane lines. Each algorithm was trained separately for 1600 episodes in Town3, and the algorithm's two metrics of average reward for episodes and average reward for single steps were counted. The episode average reward is the average of the cumulative rewards of the 10 episodes and used to evaluate the task learning; the average single-step reward records the actual number of steps interacting with each step, and the average single-step reward is obtained to further evaluate the goodness of the model through the single-step reward.

### 4.2.1. Analysis of Training Results

The training results of the episode average the reward of the algorithm are shown in Figure 5, and the single-step average reward is shown in Figure 6. Each curve stabilizes after rising, showing that the algorithm reaches convergence after exploratory learning. It can also be seen that the episode average reward and single-step average reward of the CCMR-PPO algorithm after convergence are higher than CCMR-DDPG, PPO, and DDPG, indicating that the performance is optimal after convergence, and the reward value is significantly improved relative to the original PPO and DDPG algorithms. Similarly, the effect of adding the improved parts of this paper to the DDPG algorithm also shows a significant improvement, indicating that the improvements proposed in this paper are effective. However, the CCMR-PPO algorithm converges more slowly than the DDPG algorithm, completing convergence at 300 episodes.

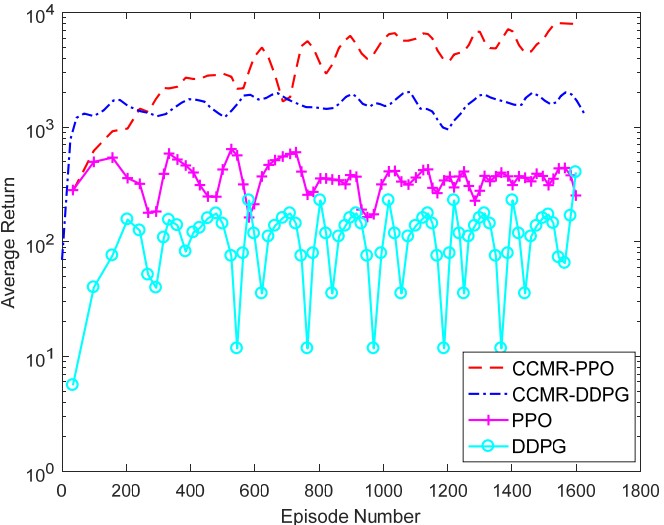

**Figure 5.** Episode average reward.

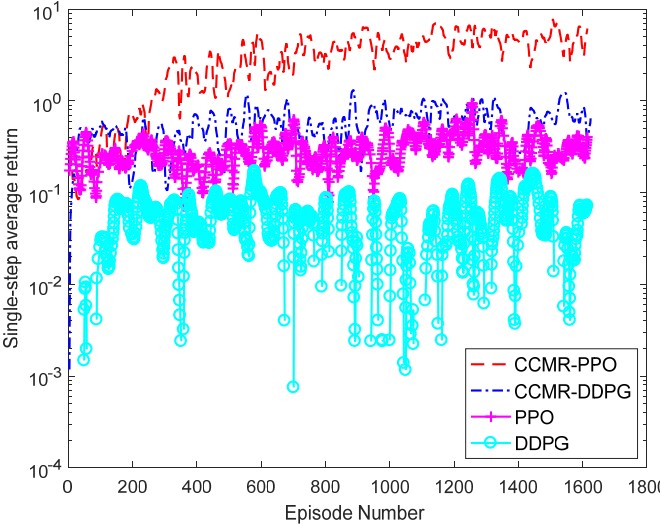

**Figure 6.** Single-step average reward.

### 4.2.2. Analysis of Test Results

The models trained by the four algorithms were placed on the Town3 where 100 other vehicles driving according to the rules were randomly placed. The test was conducted for five scenarios: driving in a straight line, turning on a curve, crossing the crossroad, turning at T-junction, and through the roundabout, and three evaluation indexes were set according

to the number of tasks completed (without triggering the termination condition) (CT), the number of times the line is crossed (CL), and the number of collisions that occurred (CO). To ensure the reliability of the experiment, 10 tests were conducted for each task, and the test results are shown in Table 4. CCMR-PPO can complete each task better, and all the indexes exceed the CCMR-DDPG, PPO, and DDPG comparison algorithms, and the number of squeezes is reduced by 24% relative to PPO, and the number of completed tasks increased by 54%. The CCMR-DDPG, which transposes the ideas of this paper to the DDPG algorithm, also performs better, and all performance indexes are improved. The original DDPG algorithm performs poorly, and the vehicle swings at a large angle while driving, which does not meet the requirements of comfort and stability, making it difficult to complete task scenarios with large turning angles such as turnings and roundabouts.

**Table 4.** Test results of the four algorithms.

| Algorithms | Straight-Road | | | Curve-Road | | | Crossroad | | | T-Junction | | | Roundabout | | |
|---|---|---|---|---|---|---|---|---|---|---|---|---|---|---|---|
| | CT | CL | CO | CT | CL | CO | CT | CL | CO | CT | CL | CO | CT | CL | CO |
| CCMR-PPO | 10 | 0 | 0 | 10 | 0 | 0 | 10 | 1 | 0 | 6 | 5 | 4 | 10 | 2 | 0 |
| CCMR-DDPG | 10 | 3 | 0 | 8 | 5 | 1 | 6 | 2 | 2 | 4 | 4 | 3 | 8 | 5 | 0 |
| cPPO | 6 | 3 | 4 | 4 | 3 | 0 | 5 | 3 | 3 | 2 | 6 | 2 | 2 | 5 | 2 |
| DDPG | 2 | 10 | 0 | 0 | 10 | 0 | 3 | 10 | 2 | 0 | 10 | 0 | 1 | 10 | 0 |

*4.3. Ablation Study*

4.3.1. Analysis of Training Results

To verify the effectiveness of the CoordConv network described and to demonstrate the actual impact that the network has on the algorithm, statistics are presented on the episode average reward and the single-step average reward with and without joining the network. The average reward results of the algorithm rounds are shown in Figure 7. From the results, the algorithm without the use of the CoordConv network has lower reward values and slower convergence. The reason for this result is that the algorithm has a better extraction effect on the features of the state input; the improvement proves the effectiveness of the CoordConv network. The single-step average reward of the algorithm is shown in Figure 8. The single-step average reward that the algorithm produces without joining the CoordConv network has a lower value, further proving the effectiveness of the CoordConv network.

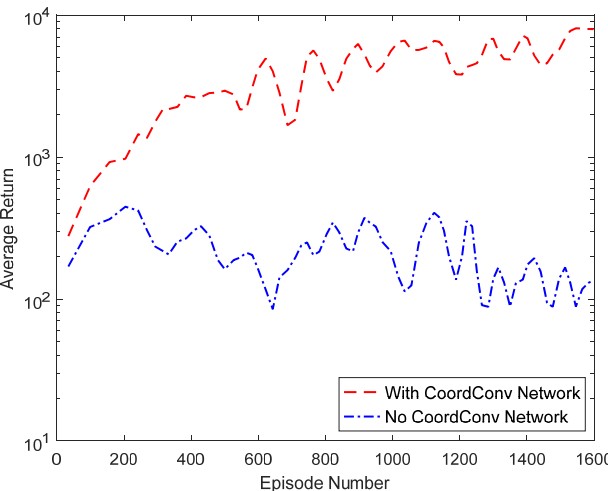

**Figure 7.** Episode average reward.

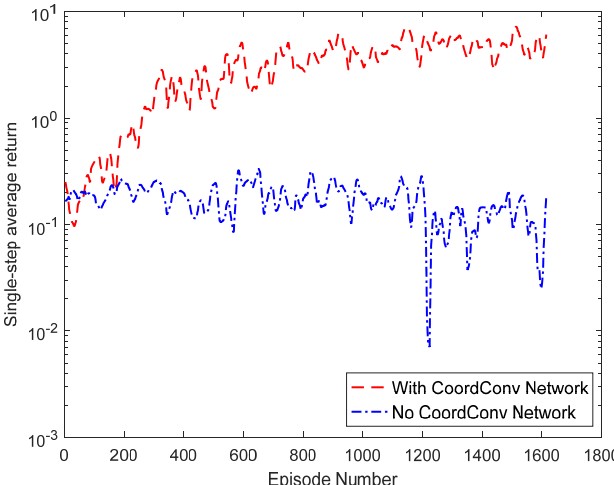

**Figure 8.** Single-step average reward.

This paper discussed the effect of the proposed reward on the algorithm model. For the proposed reward in Equation (13), the impact on the algorithm results was analyzed after removing the throttle or brake reward. As seen in Figures 9 and 10, both the average episode reward and the average single-step reward decrease significantly after removing the throttle or brake reward.

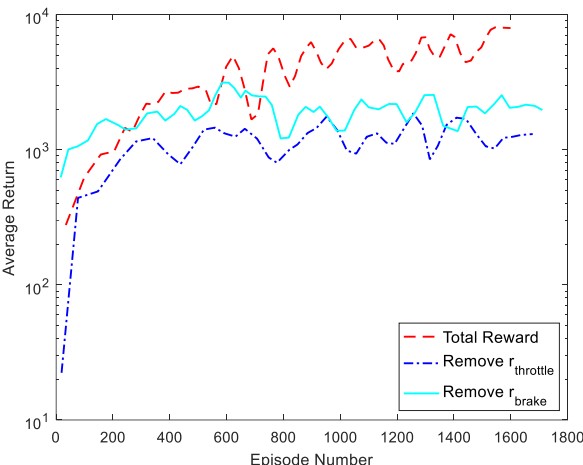

**Figure 9.** Episode average reward.

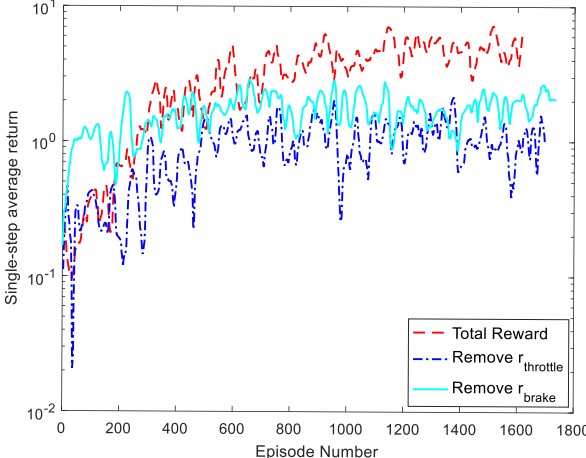

**Figure 10.** Single-step average reward.

### 4.3.2. Analysis of Test Results

For the above training model, two scenarios of roundabouts and crossroads were set up for testing, while 100 other vehicles were set up near the scenarios, and the task completion of driving in and out of the traffic circle was tested for the inner and outer circles, respectively, for the traffic circle scenario. The crossroads scenario tested the task completion of a left turn and a right turn. Each scenario task was tested for ten episodes, and three evaluation indexes were set for the number of CTs, the number of COs, and the number of CLs.

Table 5 shows that the $r_{brake}$ and $r_{throttle}$ reward terms have a greater impact on the algorithm, especially when vehicles make left and right turns at intersections, which are prone to both collisions and lane violations. Figure 11a shows a vehicle making a left turn in the middle lane of the intersection in the total reward case, and Figure 11b shows a vehicle making a right turn in the middle lane of the intersection after removing $r_{throttle}$—the vehicle crossed the line. Figure 11c shows a vehicle making a right turn in the middle lane of the intersection after removing $r_{brake}$—it collides with an oncoming vehicle.

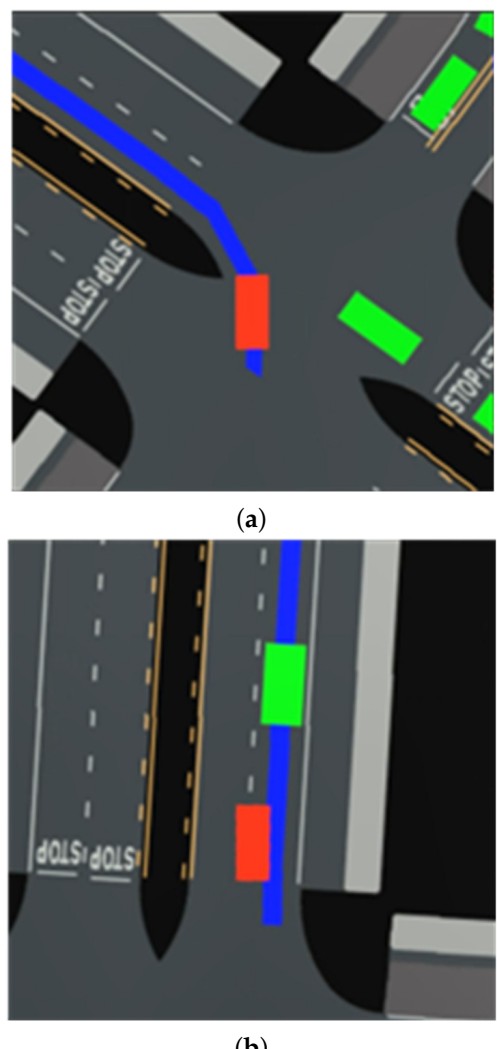

(**a**)

(**b**)

**Figure 11.** *Cont.*

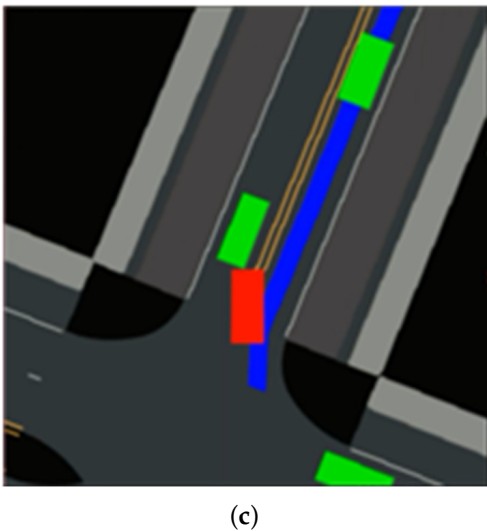

(**c**)

**Figure 11.** Comparison of ablation experiment results. (**a**) Total Return. (**b**) Remove $r_{throttle}$. (**c**) Remove $r_{brake}$.

Through the verification of simulation experimental results, the proposed algorithm in this paper can effectively improve the learning efficiency of the model, so that the network can output more rewarding vehicle driving actions, and the reward term design is scientific and effective, which improves the comfort and safety of the vehicle. To further illustrate the effect of this paper, a video link is recorded as https://pan.baidu.com/s/1ujPbbV9 mCp7mErkSO4UvBw?pwd=12a3 (accessed on 25 March 2023).

**Table 5.** Test results of the four algorithms.

| Reward | Crossroad | | | | | | Roundabout | | | | | |
| | Turn Left | | | Turn Right | | | Inner Circle | | | Outer Circle | | |
| | CT | CL | CO | CT | CL | CO | CT | CL | CO | CT | CL | CO |
|---|---|---|---|---|---|---|---|---|---|---|---|---|
| Total reward | 10 | 0 | 0 | 8 | 2 | 1 | 10 | 0 | 0 | 10 | 0 | 0 |
| Remove $r_{throttle}$ | 7 | 3 | 1 | 6 | 4 | 4 | 9 | 1 | 0 | 8 | 3 | 0 |
| Remove $r_{brake}$ | 8 | 1 | 2 | 6 | 3 | 2 | 10 | 2 | 0 | 10 | 1 | 0 |

## 5. Conclusions

This paper proposed the CCMR-PPO method based on the PPO algorithm. Firstly, the bird's-eye view data were extracted through the CoordConv network, and the extracted data and the vehicle state data were fused as the input of the algorithm, which effectively reduces the dimensionality of the state space and the exploration space of the algorithm in comparison with the forward-looking camera or lidar image data as the input. Secondly, by designing a multi-objective reward term mechanism, the algorithm can quickly obtain effective rewards to output throttle, brake, and steering control commands. The proposed algorithm was trained and verified by building a simulation environment using CARLA. Simulation results show that the CCMR-PPO method proposed in this paper has good decision-making capability, and the number of successful decisions and average rewards are improved in both traffic circle and intersection simulation environments, with a 24% reduction in the line crossed times and a 54% increase in the number of completed tasks relative to the original PPO algorithm.

Although the CCMR-PPO algorithm achieves better results at roundabouts and intersections, training and testing for challenging driving scenarios, such as more dense traffic, pedestrian involvement, and compliance with traffic rules, have not been thoroughly inves-

tigated, which is the research goal of continuing work extending the research presented in this paper.

**Author Contributions:** Funding acquisition, Y.L.; investigation, Q.S.; methodology, Q.S. and Z.L.; project administration, Q.S., H.Q. and Z.W.; resources, J.Z. and M.L.; software, Q.S. and Z.W.; supervision, Y.L. and Z.L.; writing—original draft, Q.S.; writing—review and editing, Q.S., M.L. and Z.W. All authors have read and agreed to the published version of the manuscript.

**Funding:** This work is supported in part by the National Key R&D Program under Grand 2021YFC3001300, in part by the National Natural Science Foundation of China Key Project Collaboration under Grand 61931012, in part by the Natural Science Foundation of Beijing under Grand 4222025, in part by the Science and Technique General Program of Beijing Municipal Commission of Education under Grant No. KM202011417001, and in part by the Academic Research Projects of Beijing Union University under Grant ZK10202208 and ZK90202106.

**Institutional Review Board Statement:** Not applicable.

**Informed Consent Statement:** Not applicable.

**Data Availability Statement:** The data used to support the findings of this study are available from the corresponding author upon request.

**Conflicts of Interest:** The authors declare that they have no conflict of interest.

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
