# Peer review of "Autonomous Driving Decision Control Based on Improved Proximal Policy Optimization Algorithm"

_applsci, doi:10.3390/app13116400_

Round 1

Reviewer 1 Report

This paper proposes a Coordinated Convolution Muti-Reward Proximal Policy Optimization (CCMR-PPO), which reduces the dimension of the bird’s-eye view data through the coordinated convolution network and then fuses the processed data with the vehicle state data as the input of the algorithm to optimize the state space. Experiments demonstrate the effectiveness of the proposed method. Though this paper is well written, there are also following concerns should be addressed.

1) It is suggested to divide the eq. (13) into two lines

2) As k1 is sensitive, it it suggested to provide the robust curve of k1 around the value 200. And it is suggested that the authors should provide some suggested stategies to adjust k1.

3) L197 This Paper --> This paper 

4) L342 the single-step --> The single-step

5) There should be a space before the parentheses, such as L356 completed(CT) --> completed (CT)

Author Response

Dear Editors and Reviewers,

Thank you very much for your comments and professional advice. These perspectives help to enhance the academic rigor of my manuscript. Based on your suggestions and requirements, we have made corrections and modifications to the revised manuscript. For details, please refer to the attachment.

Reviewer 2 Report

This paper proposed a novel decision-control method for autonomous driving. Overall, the paper is well organized and the results are convincing. The methodology is novel. However, I do have minor comments that may help to improve the paper.

1) Please provide more information on the training process in terms of what hardware have you used and how long it will take to make the model converge.

2) Besides reinforcement learning, there is also a lot of work based on pid and mpc. In order to ensure the integrity of the introduction of related work, the following work should be included in the introduction: a systematic survey of control techniques and applications: from autonomous vehicles to connected and automated vehicles; planning and tracking control of full drive-by-wire electric vehicles in unstructured scenario.

3) How to obtain the vehicle states such as longitudinal and lateral velocity is challenging. There are some interesting work focusing on this issue: autonomous vehicle kinematics and dynamics synthesis for sideslip angle estimation based on consensus kalman filter, improved vehicle localization using on-board sensors and vehicle lateral velocity, automated vehicle sideslip angle estimation considering signal measurement characteristic. It would be meaningful to discuss these work in the introduction.

4) Convolution network is popular to extract the spatial feature. And it also develops fast in the field of autonomous driving. Thus some related work should be included in the introduction: yolov5-tassel: detecting tassels in rgb uav imagery with improved yolov5 based on transfer learning. automated driving systems data acquisition and processing platform.

5) Equation 13 should be optimized as it is too long in one column.

Author Response

(The authors gave the same response as above.)

Reviewer 3 Report

This paper  proposes a modification to the standard PPO algorithm in RL, which is based on using inputs from a bird's eye function. The name given is Coordinated Convolution Muti-Reward Proximal Policy Optimization (CCMR-PPO). The paper then empirically compares  the new approach with PPO and DDPG using the CARLA driving simulator.

There is no theoretical analysis, so the paper rests entirely on the quality of its empirical evaluation. The results indicate that CCMR-PPO out-performs the baseline algorithms. However, the empirical analysis needs to be improved to consider this article as publishable in a journal. At present, this paper would be acceptable in a mid-tier conference, but probably not at a top-tier learning conference.

There are many limitations of the current empirical analysis.

1. Better evaluation of the reward function in sec. 2.3. (Reward Function)

    --Many parameters in the reward function, and you assume it is additive.
    --you must test the sensitivity of performance to reward function parameters (k_i) and the rewards (r_collison, etc.). Hyper-parameter optimization is the typical method, followed by a careful sensitivity analysis.
    For example, why have you chosen a Negative exponential function for r_line? Does this matter---what if you used a Gaussian?

2. Network architecture
    ---no evidence of Hyper-parameter optimization

3. RGB bird’s-eye view
    ---no evidence of Hyper-parameter optimization
    ---why was the down-sampled representation selected?

Other issues:

1. In sec. 4, you claim "which effectively reduces the .... exploration space of the algorithm."
    --please provide proper validation of this claim

2. Would like to see a more detailed analysis of "the CCMR-PPO algorithm converges slower compare to the DDPG algorithm, completing convergence at 300 episodes"
    ---what is the tradeoff of convergence vs. accuracy change?

Author Response

(The authors gave the same response as above.)

Reviewer 4 Report

1. The authors mentioned "vehicle’s acc", but it is unclear the term acc. Whether it is Adaptive cruise control, then it should be referred to as ACC

2. It will be useful if the authors could provide the different k1 value results to clearly understand the reason behind the k1 value being set to 200.

3. Provide sufficient results to select the correct value for coefficients (k1 to k8)

4. Table 4. shows that the proposed algorithm only gives better results in 2 scenarios. Compared to CCMR-PPO, it seems CCMR-DDPG gives better results. The authors are suggested to provide justification for this case.

Author Response

Dear Reviewers,

Thank you for your thorough review of our manuscript and for providing us with valuable feedback. We appreciate your efforts in reviewing our work and hope that our paper will contribute to the field of Autonomous Driving Decision-making and Control. Based on your suggestions and requirements, we have made corrections and modifications to the revised manuscript. For details, please refer to the following:

1.The authors mentioned "vehicle’s acc", but it is unclear the term acc. Whether it is Adaptive cruise control, then it should be referred to as ACC

The paper has been revised. The acc term and steer in "vehicle's acc" are the control variables in the paper, and in order not to create ambiguity, this article re-modifies the description of the sentence in L7 and L88 as follows:

The control commands acc (acc represents throttle and brake) and steer of the vehicle are used as the output of the algorithm.

Finally, the algorithm controls the vehicle’s acc and steer commands by making decision outputs from the reward values, called the Coordinate Convolution Multi-Reward function under Proximal Policy Optimization (CCMR-PPO)

2.It will be useful if the authors could provide the different k1 value results to clearly understand the reason behind the k1 value being set to 200.

In the vehicle-dense scenario, I re-evaluated the effect of different values of k1 on the results of the experiment, as shown in the figure below. The different lines in the graph indicate different values of k1 taking, where the red line is the bonus value for k1 taking 200, which increases with the number of rounds. Compared to k1 taking other values, a better reward value can be obtained when k1 takes 200, so the value of k1 is set to 200 in this paper. k1 is taken in a way that is affected by the obstacles around it and can be modified according to one's environment, and this paper focuses more on this combination of rewards to provide a reference for the design of rewards for those who follow.

Figure1

3.Provide sufficient results to select the correct value for coefficients (k1 to k8)

Thank you for your comments. The values of k1-k8 can be modified differently for different scenarios, and the parameters are selected after tuning according to the training scenarios. The reward items designed in this paper refer to the award item design of the paper [30], and at the same time add and change the correlation coefficient. Moreover, reader can adjust the parameters according to their own scenarios, and the specific adjustment method has been annotated in the paper such as L189-191, L194-196. I hope that the reward items designed in my paper can provide readers with reference. In subsequent studies, we will use other parameter optimization methods to tune them.

L189-191

This paper uses 50 as a ruler to test the decision-making control effect of vehicles in traffic flow dense and non-intensive, and the experimental results show that the value of k1 is set between 150 and 300, so the value of k1 is set to 200.

L194-196

The coefficient k2 is set to 1. If the weight is set too large, the vehicle speed will increase rapidly and cause a collision. If it is set too small, the vehicle cannot move forward, as shown in Equation (15).

The modifications to this article L173-176 are as follows:

In order to allow the agent to learn an excellent driving strategy, measure the quality of the actions performed by the agent, and solve the sparse reward problem of the algorithm, the reference paper [30] of this paper considers a total of 6 environmental variables and state factors: collision (rcollision), longitudinal speed (rspeed_lon,rfast), lane departure(rout,rline), lateral speed(rlat), brake(rbrake), and throttle(rthrottle). The last term c is a small constant set to -0.1, which was used to penalize the ego vehicle for stopping still. The design of the reward function needs to encourage the vehicle to move forward along the lane while taking care that the change in action output is as smooth as possible. This paper splits the targets according to their credit assignment, setting a scale of 50 parameter adjustment for the less frequent cases of the initial state such as the collision reward term rcollision, and a scale of 5 parameter adjustment for rfast. For the initial frequently occurring cases such as rout, rbrake and rthrottle set a parameter adjustment with a scale of 1. For the other continuously changing parameters rspeed_lon, rline and rlat set a parameter adjustment with a scale of 0.1. Finally this paper obtained the parameters in the paper and after several tests the reward function is as shown in Equation (13)

4.Table 4. shows that the proposed algorithm only gives better results in 2 scenarios. Compared to CCMR-PPO, it seems CCMR-DDPG gives better results. The authors are suggested to provide justification for this case.

I am very sorry that the paper has provided incorrect data and caused you distress. I did not check for errors here after the translation of the paper, the data in the paper before translation is shown in Figure 2 below and the translated paper is shown in Figure 3. I am very sorry for writing the wrong data. Based on your comments, we have checked the data in other tables and images and no other errors were found.

Figure 2

Figure3

Kind regards,

Qingpeng Song

Round 2

Reviewer 1 Report

The authors have addressed my concerns well.

Reviewer 2 Report

I find that the authors have put considerable effort into addressing the comments of the reviewers. As a result, the paper is very much improved, and I have no problem recommending it for publication.

Reviewer 4 Report

I appreciate that the authors have updated the manuscript to a greater extent, however, I would like to suggest the authors do extensive proofreading and grammatical corrections.

For example, the abstract itself has a spelling mistake in the fourth line "this paper proposes a Coordinated Convolution Muti-Reward Proximal Policy Optimization (CCMR-PPO)." Kindly check whether it is multi-reward or muti-reward.

Similarly, go through the entire manuscript and avoid such mistakes.

Author Response

Dear Reviewer,

Thank you very much for your comments and professional advice. These perspectives help to enhance the academic rigor of my manuscript. Based on your suggestions and requirements, we have made corrections and modifications to the revised manuscript. For details, please refer to the following:

For example, the abstract itself has a spelling mistake in the fourth line "this paper proposes a Coordinated Convolution Muti-Reward Proximal Policy Optimization (CCMR-PPO)." Kindly check whether it is multi-reward or muti-reward.

Similarly, go through the entire manuscript and avoid such mistakes.

We replaced “Muti-Reward” with ”Multi-Reward” in L4, and the details are as follows:

In order to solve the problem of high dimensional state space and sparse reward in autonomous driving decision control in this environment, this paper proposes a Coordinated Convolution Multi-Reward Proximal Policy Optimization (CCMR-PPO).

In addition, we re-read the paper and made modifications to better express its meaning and facilitate reader understanding. The details are as follows:

We replaced ”discount” with ”discounted” in L104, and the details are as follows:

which usually uses the cumulative discounted reward to define the state reward at moment t, γ is the discount factor, as shown in Equation (2).

We replaced “strategy” with “policy” in L110 and added cumulative. The details are as follows:

The goal of reinforcement learning is to learn an optimal policy π∗that maximizes the expected cumulative reward for all states, i.e., the goal of reinforcement learning is as shown in Equation (3).

We added ”discounted” in L113. The details are as follows:

Under policy π, the value of state s is denoted as Vπ(s), which represents the cumulative discounted reward brought by executing policy π from state s.

We added ”discounted” in L116. The details are as follows:

Similarly, under policy π, the value of action a taken for state s is Qπ(s, a), which represents the cumulative discounted reward brought by following policy π after performing action a from state s.

We replaced ”strategy” with ”policy” in L129. The details are as follows:

Where the ratio of the old to the new policy is denoted as shown in Equation(10).

We replaced ”Network parameters θ the update method can be expressed as shown in Equation (7) ”with ”The update method for network parameters θ can be expressed as shown in Equation(7) ” in L122-123. The details are as follows:

The update method for network parameters θ can be expressed as shown in Equation(7)

We replaced ”enhance” with ”improve” in L134. The details are as follows:

In the process of parameter updating, the PPO algorithm uses truncation to limit the update of new policies, to avoid the problem of too-large differences in policies and improve the generalization of the algorithm.

We replaced ”above” with ”higher than” in L349. The details are as follows:

It can also be seen that the episode average reward and single-step average reward of the CCMR-PPO algorithm after convergence are higher than CCMR-DDPG, PPO, and DDPG, indicating that the performance is optimal after convergence, and the reward value is significantly improved relative to the original PPO and DDPG algorithms.

I would like to request your approval for these changes, which I believe will enhance the clarity, coherence, quality, and relevance of my research.

Round 3

Reviewer 4 Report

My comments were addressed